# Double adversarial domain adaptation for whole-slide-image classification

**Yuchen Yang**[1]                                                    yy17@ualberta.ca

**Amir Akbarnejad**[1]                                              ah8@ualberta.ca

**Nilanjan Ray**[1]                                                    nray1@ualberta.ca

**Gilbert Bigras**[1,2]                                    gilbertbigras@gmail.com

[1] *Department of Computing Science, University of Alberta*

[2] *Department of Laboratory Medicine and Pathology, University of Alberta*

## Abstract

Image classification on whole-slide-image (WSI) is a challenging task. A previous work based on Fisher vector encoding provided a novel end-to-end pipeline with promising accuracy and computational efficiency. However, the pipeline suffers from an accuracy drop due to domain shift. This poses a limitation on the practical use of the pipeline especially when the diagnoses of WSIs are hard to obtain. This paper aims for a solution to mitigate the accuracy drop by using an unsupervised domain adaptation approach. We propose to insert the domain classifiers into the pipeline in two stages to align the features during training. We evaluate accuracy by calculating the confusion matrices before and after the adaptation on two datasets. We demonstrate that placing domain classifiers in different stages will boost accuracy.

**Keywords:** whole-slide-image, classification, domain adaptation, deep learning.

## 1. Introduction

Deep learning for classification of whole-slide-image (WSI) is challenging. The challenge comes from its high resolution and sparsely scattered diagnostic information. Recently, a new end-to-end embedding method - Deep Fisher vector coding (DFVC) (Akbarnejad et al., 2021) is proposed. In this paper, we enhance DFVC pipeline to cope with data distribution shift.

Data distribution shift often results in an accuracy drop when a DNN model is trained on one WSI dataset and tested on another. Potential causes could be different staining processes by different institutions, WSI scanned in different periods or machines, and so on. Also, the cost of labeling WSIs is high, diagnoses such as HER2 or gleason scores are expensive to obtain. In this paper, we dedicate to utilize the unsupervised domain adaptation (UDA) approach to mitigate the accuracy drop among WSI datasets.

Domain adaptation for medical imaging is previously discussed in works such as (Ren et al., 2018). However, previous attempts only focus on adapting image patches extracted from the WSI to achieve higher patch-based accuracy. Their approaches are limited by the patch-based classification pipelines without consideration on adapting and classifying the WSI as a whole. This paper is established on the DFVC pipeline for WSI classification. We propose a UDA solution that integrates the original pipeline with domain classifiers in two stages to minimize the accuracy decrease. Comparison of a model without adaptation, adapted models, and an oracle model is demonstrated to show the effectiveness of our solution.

## 2. Methods

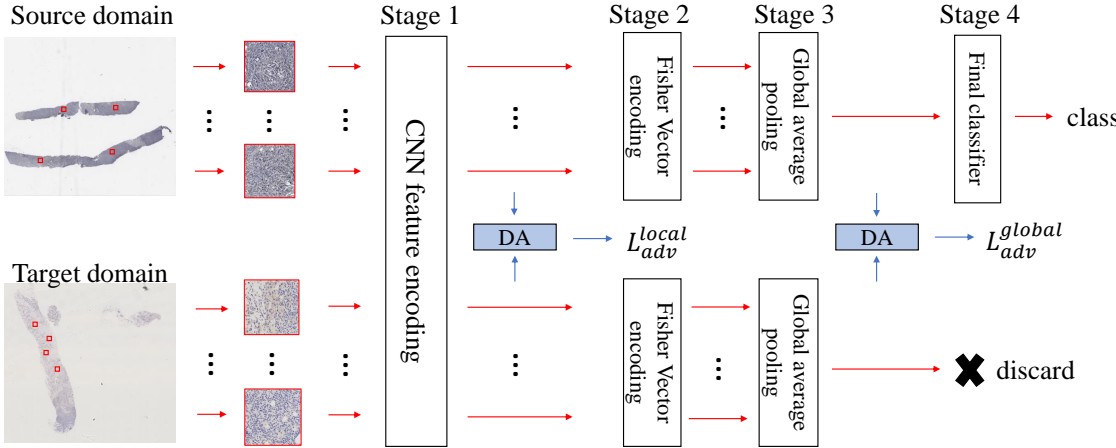

Figure 1: Overview of dual stages adaptation for WSI classification.

We show the our integrated UDA solution in Fig.1. First, the WSIs of both source and target domains are randomly sampled and augmented which follows the original DFVC pipeline, and then fed into a CNN for feature encoding. Afterward, we forward the features to a domain classifier. This domain classifier works with the patch-wise features and is responsible for adapting local distribution shifts on patches from different domains. The domain classifier outputs the two domain labels - source and target and is trained with binary cross-entropy loss. A gradient reverse layer is attached on the top of the domain classifier to enable adversarial training to adapt features from the two domains.

Besides forwarding the features to the local domain adaptation part, both source and target CNN encoded features from stage one are passed to the next stages. The features are further possessed by the Fisher vector encoding stage and then the global average pooling stage. The global average pooling stage aggregates the individual features so that each WSI is represented by a single vector. We insert the domain classifier to this stage to align the aggregated features. This domain classifier adapts the feature distribution shift of the entire WSI. The structure of the domain discriminator in this stage shares the same configuration as in the first stage with adjusted feature size.

During the training stage, the cross-entropy loss from the original pipeline and adversarial loss is combined with a balance parameter $\lambda = 0.1$ to update the entire model. The entire loss calculation of the proposed method can be estimated as:

$$Loss = L_{CE} + \lambda(L_{adv}^{local} + L_{adv}^{global}), \tag{1}$$

$L_{adv}^{local}$ and $L_{adv}^{global}$ represent the adversarial loss terms of the two domain classifiers attached after the first (local) and the third (global) stage. The domain classifiers are only attached during training and are discarded during the testing. [1]

---

1. An implementation can be found in https://github.com/yuchen2580/double_adaptation_WSI

## 3. Experiments and conclusion

We test our method on two HER2 IHC breast tissue datasets. One has 250 WSIs collected from Alberta Cross Cancer Institution (CCI dataset). The other has 52 WSIs collected from Warwick HER2 challenge (Qaiser et al., 2018) (Warwick dataset). We split the Warwick dataset in half to create a train set (24 WSIs) and a test set (28 WSIs). Ratios of categories are kept the same. For adaptation, we train the network with CCI data with label and Warwick train set without label, and evaluate the model on Warwick test set.

The classification goal is to predict 4 categories of HER2 scores (0,+1,+2,+3) in (Qaiser et al., 2018) for each WSI. Table 1 (c) and (d) show that both adaptation on local stage and global stage can help increase the accuracy. Compared to the global stage adaptation, the local stage adaptation has a better influence on all categories in the matrix. But global stage adaptation provides better separation between category 0 and category 3+. From the result of Table 1(e), the double stage adaptation provides the best accuracy and confusion matrix compared to the single stage adaptation in (c) and (d). Note that the training set of the CCI data is significantly large compared to the Warwick dataset, the increased accuracy in this experiment also indicates that our solution can apply to the scenario where the model could be trained in a bigger dataset and adapt to a smaller dataset elsewhere.

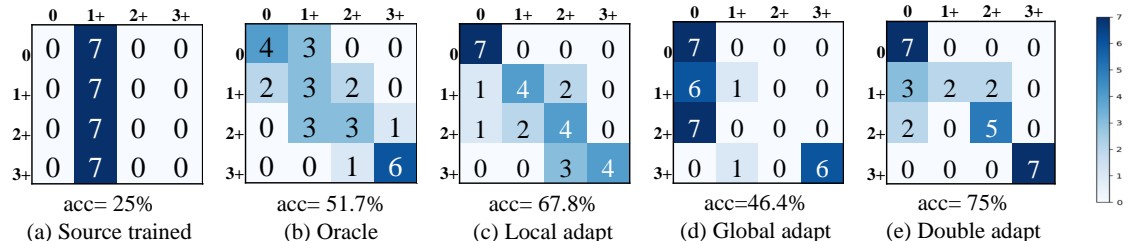

Table 1: Comparisons of confusion matrices.

In conclusion, this paper focuses on the domain shift problem that exists in WSI classification task. Built on a previous pipeline, we propose to integrate the domain classifiers into two stages to cover local and global distribution shifts. The adapted model from a big HER2 dataset to a small one, shows a significant accuracy boost in the experiment.

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
