# OpenReview forum: "Double adversarial domain adaptation for whole-slide-imageclassification"
_MIDL.io/2021/Conference/Short — MIDL 2021 Poster_

### Official Review · Reviewer_ch5y · 2021-04-20

**Confidence:** 4
**Final Rating:** 3

**Summary:**

The paper proposes to use two adversarial domain discriminators to train domain-invariant WSI Fisher vector encodings based classification models. The empirical results show that the use of both local and global domain discriminators improves the performance of the target domain. In term of novelty, the use of more than one domain discriminators has been studied before and the application in WSI classication is novel.

**Strengths:**

1. The application of domain adaptation on the WSI classication task is an important research problem.
2. The paper presents empirical results that demonstrate that the local and global domain discriminators improve the performance of the target domain.

**Weaknesses:**

I do not think it is novel to use multiple domain discriminators in the domain adaptation literature. Please refer to the following papers:

Label Efficient Learning of Transferable Representations across Domains and Tasks, NIPS2017
Multi-Adversarial Domain Adaptation, AAAI2018

**Deanonymize Review:**

no

**Detailed Comments:**

I'm not able to find the reference (Akbarnejad et al., 2021) online and am wondering how the Fisher vectors work exactly.

**Justification Of The Rating:**

I appreciate the applied contributions of this manuscript in applying domain adaptation to WSI classication, although I believe the methodological contribution of "adversarial domain adaptation" is highly incremental.

**Paper Type:**

validation/application paper

**Special Issue:**

no

---

### Official Review · Reviewer_P43z · 2021-04-29

**Confidence:** 4
**Final Rating:** 3

**Summary:**

This work is built up on the pipeline of the recently published paper, DFVC [Amir Akbarnejad et al.],  Distinct from existing works that apply domain adaption (DA) only on patch level, this work extends to apply DA on both patch level and slide level of whole slide images. The confusion matrix provided shows the benefit of this strategy.

**Strengths:**

- Well written and easy to follow.
- The paper is well motivated and the extension of DA to slide-level is intuitive.
- The idea of applying DA to both patch-level and slide-level is novel and interesting.

**Weaknesses:**

-  If more performance metrics can be provided, the paper will be more convincing. However, this is OK given it is a short paper.
- The experimental setting is not clear enough.
- The figure needs to be improved to make the idea more easier to be understood.

**Deanonymize Review:**

no

**Detailed Comments:**

- Please compact the paper to 3 pages including references.
- The experimental setting is a bit unclear. Is that the whole CCI dataset (with labels and being included in the L_ce) along with half the Warwick dataset (without being included in  the L_ce) were used for training, and the remaining 28 slides from Warwick were for testing ?


**Justification Of The Rating:**

This paper presents an interesting idea that extends the way of using DA for analyzing whole slide images. Despite a few drawbacks, I think the work is of interest to the community and I expect a thorough validation in its full version paper.

**Paper Type:**

methodological development

**Special Issue:**

no

---

### Meta-Review · Program_Chairs · 2021-05-06

**Recommendation:** Accept (Poster)
**Confidence:** 5

**Metareview:**

Both reviewers support acceptance. Authors are suggested to address reviewer comments in final version.

---

### Decision · Program_Chairs · 2021-05-11

Accept (Poster)